# Evaluation of Pre-Pandemic Trivalent COBRA HA Vaccine in Mice Pre-Immune to Historical H1N1 and H3N2 Influenza Viruses

**DOI:** 10.3390/v15010203

**Published:** 2023-01-11

**Authors:** Pan Ge, Ted M. Ross

**Affiliations:** 1Center for Vaccines and Immunology, University of Georgia, Athens, GA 30602, USA; 2Florida Research and Innovation Center, Cleveland Clinic, Port Saint Lucie, FL 34987, USA; 3Department of Infectious Diseases, University of Georgia, Athens, GA 30602, USA; 4Department of Infection Biology, Lehner Research Institute, Cleveland Clinic, Cleveland, OH 44195, USA

**Keywords:** influenza, hemagglutinin vaccine, trivalent, COBRA, pre-immunity, imprinting, broadly reactive antibody, stem antibody, Th1/Th2

## Abstract

Initial exposure to influenza virus(es) during early childhood produces protective antibodies that may be recalled following future exposure to subsequent viral infections or vaccinations. Most influenza vaccine research studies use immunologically naïve animal models to assess vaccine effectiveness. However, most people have an extensive influenza immune history, with memory cells produced by viruses or vaccines representing multiple influenza viruses. In this study, we explored the effect influenza seasonal virus-induced immunity has on pre-pandemic influenza virus vaccination. The mice that were pre-immune to historical H1N1 and H3N2 seasonal influenza viruses were vaccinated with adjuvanted pre-pandemic (H2, H5, and H7) HA-based computationally optimized broadly reactive antigen (COBRA) vaccines, and were fully protected from lethal challenge, whereas the mock-vaccinated mice, with or without pre-immunity, were not protected from morbidity or mortality. Detectable antibody titers were present in the pre-immune mice vaccinated with a single dose of vaccine, but not in the immunologically naïve mice. The mice vaccinated twice with the trivalent COBRA HA vaccine had similar antibody titers regardless of their pre-immune status. Overall, seasonal pre-immunity did not interfere with the immune responses elicited by pre-pandemic COBRA HA vaccines or the protection against pre-pandemic viruses.

## 1. Introduction

Avian influenza viruses naturally spread among waterfowls, including domestic and wild bird populations [1]. Migratory birds can carry the viruses over long distances to new locations [2,3]. Most often, infected migratory birds, such as ducks and geese, interact with domesticated chickens and turkeys, on farms. While these migratory birds usually do not show signs of disease, domesticated chickens and turkeys will have weight loss with severe dehydration and signs of illness. Close contact between people and sick birds facilitates the transmission of the virus to humans, that, depending on the subtype, leads to severe disease and death [4,5,6].

Over the past 150 years, influenza A and influenza B viruses have caused millions of human infections during annual seasonal epidemics. Occasionally, there are spikes in infections and death as a result of the introduction of a novel subtype or strain of influenza A viruses into the human population that leads to a worldwide pandemic. Since influenza A viruses can infect a broad range of hosts that can transmit them to humans, the introduction of a zoonotic influenza virus that can sustain human-to-human transmission is a threat of real concern to human health [7].

Since 1918, only three influenza A subtypes (H1N1, H2N2, and H3N2) have circulated in the human population [8]. The introduction of each of these subtypes resulted in four recorded pandemics [7]. Among the subtypes that are of high risk to spark a new pandemic are the viruses in the H7N9, H2N2, and H5Nx subtypes [9,10,11]. The novel A/H7N9 avian-origin viruses caused 1568 laboratory-confirmed cases and 615 deaths in seven waves of the outbreaks in 2013–2017 [12]. These viruses are of high concern because they spread quickly and most patients become severely ill, compared to diseases caused by other seasonal influenza A viruses [13]. Influenza viruses of the H5 subtypes also have a risk of causing a pandemic. The first human H5N1 infections were recorded in 1997 and caused at least 865 human infections and 456 deaths from 2003 to October 2022 [14,15]. Additionally, H5N1, H5N2, and H5N6 influenza viruses have spread across the world, causing outbreaks with severe morbidity and mortality in domesticated poultry in several countries in Asia, Europe, Africa, and the Americas [16,17]. These viruses are continuing to evolve and infect people. In December 2020, the first H5N8 human infection was recorded on a poultry farm in Russia [18]. Therefore, both H7N9 and H5N1 avian influenza viruses are of particular concern, not only because have they caused disastrous consequences to poultry markets, but also because they have led to severe human infections and deaths [19]. To date, H5 and H7 influenza viruses have not caused a pandemic, but they are at risk of doing so, as they have a ~40–60% fatality rate [20]. The pandemic potential of these viruses should not be underestimated; there is an urgent need to develop pre-pandemic vaccines [21].

Even though H1N1 and H3N2 influenza viruses are still co-circulating in people today, H2N2 influenza viruses are still at high risk to cause a new pandemic [11]. Influenza viruses of the H2N2 subtype caused the pandemic in 1957. H2N2 influenza viruses circulated in people for ~11 years when there was a shift to a novel H3N2 influenza virus subtype that resulted in the 1968 pandemic. Therefore, people born after 1968 have little to no immunity to H2Nx influenza viruses [22]. Even though H2 influenza viruses circulate in avian and occasionally swine species, some of these viruses can cause severe illness and death in people [23,24]. In 2006, the H2N3 swine influenza viruses (A/swine/Missouri/2124514/2006 or SW/MO/06) were isolated from pigs in two farms in the United States [25]. Since avian/swine H2 viral hemagglutinins are structurally similar to the human HA of pandemic H2 viruses, the adaptation of these viruses to the human host is plausible. Overall, if H2Nx viruses re-emerge and begin efficiently circulating in the human population, most people will be at high risk for severe morbidity or high rates of mortality due to the lack of pre-existing anti-H2 immunity [10].

To develop broadly reactive influenza virus vaccines, a methodology was used to optimize the antigen design, termed the computationally optimized broadly reactive antigen (COBRA). HA immunogens were successfully designed and tested for the H1, H2, H3, and H5 HA subtypes, as well as the N1 and N2 NA subtypes [26,27,28,29,30,31,32,33,34]. This methodology utilizes multiple rounds of consensus sequence layering that generate HA or NA antigens that are able to elicit broadly reactive immune responses to protect against both seasonal and pre-pandemic strains [26,27,31,32,33,34]. In this study, a trivalent pre-pandemic rHA vaccine was tested in mice with a pre-existing immunity to human seasonal H1N1 and H3N2 influenza A viruses to determine the effectiveness of this adjuvant-formulated vaccine at eliciting broadly protective immune responses against pre-pandemic influenza virus challenge.

## 2. Materials and Methods

### 2.1. Design and Production of COBRA HA Proteins

The computationally optimized broadly reactive antigen methodology was used to design influenza HA proteins representing the H2, H5, and H7 subtypes [27,31]. Briefly, full length HA protein amino acid sequences representing H2, H5, and H7 subtypes were downloaded from the Global Initiative on Sharing Avian Influenza Data (GISAID) and National Center for Biotechnology Information (NCBI) Databases as previously described [27,31]. Following multi-layering consensus building [26,30], the final consensus sequences for each subtype were selected and the soluble HA gene sequences representing amino acids 1 to 566 were synthesized (Genewiz, South Plainfield, NJ, USA) without the transmembrane domain and cytoplasmic tail and cloned into the Zeo+ pcDNA 3.1 vector. Each HA gene was cloned in frame with a T4 fold-on domain, an Avitag sequence, and a histidine (6X) tag for purification. The final protein concentration was determined using a bicinchoninic acid assay (BCA). The soluble H2 HA was designated Z1, the soluble H5 sequence was designated IAN8, and the soluble H7 HA was designated Q6. The Z1 and IAN8 COBRA HA antigens have previously been described [27,31], but the H7 COBRA, Q6, was designed using both North American and Eurasian strains to incorporate H7 HA sequences isolated from all the species during the time period from 1970 to 2018 (Dr. Hyesun Jang, personal communication).

### 2.2. In Vitro Protein Expression and Virus-like Particle Production

Total of thirteen HA sequences were expressed on the surface of virus-like particles (VLPs). Briefly, the full length H2, H5, and H7 HA amino acid sequences were subjected to codon optimization and inserted into pTR600 expression vector. The human embryonic kidney (HEK) 293T cells transiently transfected (Lipofectamine^TM^ 3000, Thermo Fisher Scientific, Waltham, MA, USA) with plasmids expressing Has, human immunodeficiency virus-1 (HIV-1) Gag, and NA (NA genes from 1918 H1N1 virus for H2 VLPs, NA genes from A/Thailand/1(KAN-1)/2004 (H5N1) for H5 and H7 VLPs. The cells were incubated at 37 °C, and supernatant was harvested, centrifuged, and filtrated. Filtered supernatant was further purified via ultracentrifuge (27,000× *g* through 20% glycerol) for 4 h at 4 °C. Pellets were resuspended in PBS and stored in aliquots at −80 °C.

### 2.3. Vaccinations

All animal procedures were approved by the University of Georgia Institutional Animal Care and Use Committee (IACUC) #A2018 06-018-Y3-A18 and performed in accordance with institutional guideline. Eight-week-old female DBA2/J mice (The Jackson Laboratory in Bar Harbor, ME, USA) were used for all the animal experiments. Mice in each set (3 sets in total) were randomly divided into 4 different groups (N = 15). Mice labeled, Pre-immune Mock, were intranasally infected with equal amounts of A/Singapore/6/1986 (H1N1) and A/Panama/2007/1999 (H3N2) viruses at a final concentration of 5 × 10^6^ PFU/50 μL in PBS and then mock-vaccinated with 50 μL PBS six weeks after pre-immunization. Mice labeled, Pre-immune COBRA, received the same amount of H1N1 and H3N2 viruses as above and were intramuscularly vaccinated twice with 9 ug total COBRA recombinant HA in PBS formulated with AddaVax^TM^ (InvivoGen, San Diego, CA, USA) at an interval of four weeks; mice in Naïve COBRA group were mock pre-immunized with 50 μL PBS and vaccinated with the same trivalent vaccine described above; mice in Naïve Mock group were mock-pre-immunized and mock-vaccinated. Groups includes Pre-immune Mock, Pre-immune COBRA (vaccinated), Naïve COBRA (vaccinated), and Naïve Mock. Mice in groups 1–4 were challenged with Mall/MN/2008 virus, while groups 5–8 were challenged with Sichuan/2014 virus. Mice in group 1 and 5 were pre-immunized with Sing/86 and Pan/99 then mock-vaccinated with PBS; group 2 and 6 were pre-immunized with Sing/86 and Pan/99 then vaccinated with Z1, IAN8, Q6, 3 ug of each formulated with AddaVax^TM^; naïve mice in group 3 and 7 were vaccinated with the same vaccine mixture; naïve mice in group 4 and 8 were mock-vaccinated with PBS instead (Table 1).

Blood was collected via submandibular bleeding from each animal on day 0 prior to vaccination and days 14, 42, and 49 post-vaccination. On day 35 post-vaccination, spleens were harvested from 3 mice in each group. On day 56, mice were challenged intranasally with 2 × 10^7^ PFU/50 μL A/Mallard/Minnesota/2008 (H2N3), or 1 × 10^6^ PFU/50 μL A/Sichuan/26221/2014 (H5N6). Following challenge, mice were weighed daily and monitored for clinical signs of illness for 14 days. On day 59 and 62, lungs were collected from 3 mice per group (Figure 1). Any mouse that lost more than 25% of its original weight or had severe clinical symptoms was humanely euthanized when the clinical score was equal or larger than 3. Survival data are presented as the percentage of surviving animals at each day, compared to the initial number of animals in each group.

### 2.4. Enzyme-Linked Immunosorbent Assay (ELISA)

Anti-HA IgG serum antibodies were determined by ELISA. Briefly, 96-well flat-bottom plates were coated with 100 mL/well of 10 mg of total protein of rHA (Z1, IAN8, Q6, cH6/1-H6 head from A/Mallard/Sweden/81/2002 and H1 stalk from A/California/07/2009, cH7/3-H7 head from A/Anhui/1/2013 and H3 stalk from A/Texas/50/2012) overnight at 4 °C. Plates were washed with PBS containing 0.05% Tween 20 (PBST) and blocked for 90 min at 37 °C with 1% bovine serum albumin in PBST solution. Following a wash step, three-fold serial dilution of the mouse serum in blocking buffer was added for 90 min at 37 °C or at 4 °C for overnight. Each serum sample was tested in duplicate. After washing, goat-anti-mouse IgG horse-radish peroxidase-conjugated secondary antibodies (Southern Biotech, Birmingham, AL, USA) were added at a 1:4000 dilution, and goat-anti-mouse IgG1, IgG2a, IgG2b (Southern Biotech, Birmingham, AL, USA) or anti-mouse IgA (Southern Biotech, Birmingham, AL, USA) were added at a 1:100 dilution, for 90 min at 37 °C. After 5 times washed, 100 μL 0.1% 2,2′-azino-bis (3-ethylbenzothiaozoline-6 –sulphonic acid; ABTS) solution with 0.05% H_2_O_2_ for 15 min at 37 °C, 50 μL of 1% sodium dodecyl sulfate (SDS) solution was added to stop the colorimetric development and the absorbance was read at 414 nm using a PowerWaveXS (Biotek, Winooski, VT, USA) plate reader. The optical density (O.D.) value of the blanks was averaged then subtracted from all the values in each well, then the O.D. values of each duplicate were also averaged, and endpoint antibody titers were expressed as the highest dilution in which the O.D. value is greater than 2 times of the blanks values.

### 2.5. Hemagglutination-Inhibition (HAI) Assay

HAI assays were performed according to the WHO Manual for the laboratory diagnosis and virological surveillance of influenza in 2011 version (WHO, 2011). Briefly, serum samples were treated with receptor-destroying enzyme (RDE) (Denka Seiken, Co., Tokyo, Japan) for 16 h at 37 °C, followed by 45 min incubation at 56 °C. RDE-treated serum was two-fold serially diluted in v-bottom microtiter plates. Influenza virus or VLP in 25 mL of phosphate buffer saline (PBS) at 8 hemagglutination units (HAU)/50 μL was added to each well. The plates were gently rocked then incubated for 20 min then mixed with 0.8% turkey red blood cells (TRBC) or incubated for 30 min then added to 1% horse red blood cells (HRBC) at RT. TRBC were used for H2 and H7 viruses and HRBC were used for H5 viruses in the HAI assay. The RBCs were then allowed to incubate for 30 min for TRBC or 1 h for HRBC. HAI titer was determined to be the reciprocal dilution of the last well that contained non-agglutinated RBCs. Positive and negative serum controls were included for each plate. All mice were negative (HAI < 1:10) for pre-existing antibodies circulating human influenza viruses prior to study onset.

### 2.6. Lung Viral Titers

The right lung lobe was harvested on day 59 from each group of mice and all lobes of the lungs were collected on day 62. Briefly, lung samples (n = 3 per timepoint) were collected and flash frozen, then stored at −80 °C until processing. Following thawing on ice, lung tissues were homogenized and diluted 10X in DMEM (Thermo Fisher, Waltham, MA, USA), and supplemented with 1% penicillin-streptomycin (DMEM+P/S). Lungs were homogenized and serially diluted (100 to 105) with DMEM+P/S and overlayed onto 85~95% confluent Madin-Darby Canine Kidney (MDCK) cell layer for 1 h. Cells were washed twice with dilution solution DMEM+P/S and overlaid with 1 mg/mL TPCK trypsin and evenly mixed with 2x MEM and 1.6% agarose. After incubation for 3 days at 37 °C, the overlay was removed from each well and cells were fixed with 10% formalin for 15 min and the monolayer was stained with 1% crystal violet (Thermo Fisher, Waltham, MA, USA) for 15 min for plaque counting. Results are expressed as plaque forming unit (PFU) per lung tissue.

### 2.7. Quantification and Statistical Analysis

Data are presented as absolute mean values ± standard error of the mean (SEM). One-way ANOVA and two-way ANOVA were used to analyze the statistical differences when appropriate. A “*p*” value less than 0.05 was defined as statistically significant (*, *p* < 0.05; **, *p* < 0.01; ***, *p* < 0.001; ****, *p* < 0.0001; ns, not significant). All statistical analyses were performed using GraphPad Prism 9 software (GraphPad, San Diego, CA, USA).

## 3. Results

### 3.1. COBRA rHA Vaccinations Provide Protection against H2N3 and H5N6 Influenza Virus Challenge

To evaluate the efficacy of the trivalent COBRA rHA vaccine, mice with different pre-immune backgrounds and vaccination regimens were compared (Figure 1, Table 1). Prior to vaccination, the mice seroconverted to Sing/86 and Pan/99 following infection (Appendix A). The naïve mice infected with Mall/MN/2008 (H2N3) lost more than 20% of their original body weight by day 7 post-infection, with 50% survival, then began to gain weight (Figure 2A). The COBRA rHA-vaccinated mice that were naïve or pre-immune lost no more than 10% of their original body weight by day 5 post-infection, which was similar to the weight loss in naïve mock-vaccinated mice. One pre-immune mouse that was mock-vaccinated died from infection. All the other mice survived infection (Figure 2B). Moreover, the protection induced by pre-immunization or vaccinations was confirmed by the assessment of viral loads recovered on both day 3 and day 6 post-challenge (Figure 2C). High lung viral titers were detected at both day 3 post-challenge in naïve mice and pre-immune, non-vaccinated mice, but were undetectable in the lungs of mice in either group of the COBRA HA-vaccinated mice. Lung titers were detected on day 6 in only naïve, unvaccinated mice (Figure 2C).

Another set of DBA2/J mice were evenly separated into four different groups, and half were infected with the same Sing/86 and Pan/99 viruses to establish pre-immunity. Both pre-immune and naïve mice were divided into two groups, where half were vaccinated with the trivalent COBRA HA mixture or were mock-vaccinated. On day 56, the mice were challenged with the A/Sichuan/26221/2014 (H5N6) influenza virus. All the vaccinated mice survived the lethal challenge by the H5N6 virus with little weight loss (Figure 2D,E). In contrast, the naïve and unvaccinated mice (Naïve Mock) were not protected and all died following the H5N6 influenza virus challenge, with all pre-immune, non-vaccinated mice surviving the challenge, but losing ~15% of their original body weight. On day 7 post-challenge, the mice in both of the vaccinated groups (Pre-immune COBRA and Naïve COBRA) had little weight loss and all survived H5N8 infection with little weight loss. Similar to the H2N3 influenza viral challenge, no virus was detected in either vaccinated group on either day 3 or day 6, but high viral titers were recovered for mice in the Naïve Mock group on both day 3 and day 6 post-challenge, and lower lung titers were detected in the Pre-immune Mock group, but were no longer detectable on day 6 post-challenge (Figure 2F).

### 3.2. Trivalent COBRA rHA Vaccinations Induce High Titers of Total IgG Antibody

To assess the impact of pre-immunization and vaccinations on the total anti-HA binding antibody titers, the HA-specific serum IgG endpoint was measured in all four groups of mice. The mice vaccinated with COBRA rHA vaccines had significantly higher HA-specific antibodies against all three vaccine HA components than the mice that were not vaccinated. The pre-immune mice had a higher total IgG, with no significant differences compared to the naive mock-vaccinated mice (Figure 3A–C). Several mice that were pre-immunized and mock-vaccinated had low levels of anti-HA-specific IgG, with no differences observed compared to the mock-vaccinated naïve mice.

### 3.3. Trivalent COBRA rHA Vaccinations Induce High Hemagglutination-Inhibition (HAI) Antibody Titers

Having observed high protective efficacy in the pre-immunized mice vaccinated with the trivalent COBRA rHA vaccinations, immunological correlates were investigated. Serum samples were collected on days 42 and 49 and titrated for receptor-blocking antibodies by HAI assay against a panel of five H2 virus-like particles (VLP) representing all three clades of H2Nx viruses isolated between 1964–2016 (Figure 4A–D). The serum collected from mice that were vaccinated with the trivalent COBRA rHA all elicited antibodies (Figure 4B,C), and had average titers that were higher, but not statistically significantly so, against three of the viruses in the panel than the sera from mice that were pre-immune and vaccinated with the same COBRA HA vaccines. Although the pre-immune and naïve mice that were mock-vaccinated had different morbidity and survival following the challenge, the H2 HAI titers were statistically similar. Neither group had detectable HAI titers, which was consistent when the same serum samples were tested for the HAI titers against H5 or H7 viruses/VLPs (Figure 4A,D,E,H,I,L).

The same sera collected on days 42 and 49 were also titrated for receptor-blocking antibodies against a panel of six H5Nx viruses spanning from 2004–2020. The pre-immune and naïve mice that were vaccinated with COBRA HA all had very high HAI titers against the most recent strains, gy/Wa/14, SC/14, and Ast/20, that represent the clade 2.3.4.4. These same mice sero-converted HAI titers against VN/04, WS/05, and HB/10 (Figure 4F,G). No seroconversion was detected in the other two mock-vaccinated groups (Figure 4E,H).

Sera collected from mice had seroprotective antibodies against a panel of six H7 VLPs spanning the timeframe from 2000–2016, representing both Euro-Asian strains and North American strains. Mice in both Pre-immune COBRA and Naïve COBRA groups possessed seroprotective HAI antibodies against each VLP, no difference was found in these two groups against the panel of viruses. Similarly, there was no seroconversion in pre-immune mice or naïve mice that were mock-vaccinated (Figure 4I,L). These results indicate that H1N1/H3N2 pre-immunization was not able to elicit receptor-blocking antibodies against H2, H5, or H7 viruses or VLPs.

### 3.4. H1/H3 Pre-Immunity Contributes to Group 1 and Group 2 Anti-HA Stem IgG, but Shows Little Effect after Two COBRA HA Vaccinations

To determine if pre-immunity contributes to the elicitation of anti-HA stem antibodies following COBRA HA vaccination, serum samples were assessed for both group 1 and group 2 anti-HA antibodies (Figure 5). On day 7 after each vaccination, anti-stem reactive antibodies were detected in serum samples collected after prime (Figure 5A,C) and after boost (Figure 5B,D) vaccination. Both pre-immune COBRA and naïve COBRA HA-vaccinated mice had higher anti-group 1 stem and anti-group 2 stem IgG antibodies after the boost (Figure 5B,D). Interestingly, a single trivalent COBRA rHA vaccination in pre-immune mice elicited similar levels of anti-group 2 IgG antibodies (Figure 5C). The naïve mice vaccinated with COBRA HA proteins had significantly higher anti-group 1 IgG titers after the prime (Figure 5A) compared to the mock-vaccinated mice with or without pre-immunity. As shown in Appendix A, following the prime or boost, the mice had higher anti-group 1 IgG antibody titers. The immunodominance against group 1 HA might be due to there being two group 1 components in the vaccine but one group 2 component. These findings suggest that H1/H3 stem-derived fragments are likely to contribute to the antibody-mediated immune responses in pre-immune mice that have been mock-vaccinated.

### 3.5. COBRA HA Vaccinations Elicit Robust Th2 Biased Antibody Responses

Anti-HA IgG1 and IgG2b antibodies were elicited by each of the COBRA HA vaccines, as well as the chimeric stem HA vaccines, against their specific antigen with ~1 log lower titers against the cH7/3 chimeric HA protein (Appendix A). Overall, the anti-HA IgG2a antibodies were ~1 log lower in titer (1 × 10^4.5^) (Appendix A) and the anti-HA IgG2b titers were ~2 logs lower than the anti-HA IgG1 titers (1 × 10^5.5^) (Appendix A). The mice in both vaccinated groups, with or without pre-immunization, induced a higher IgG1/IgG2a ratio than the mice that had only been pre-immunized but not vaccinated (Figure 6). Overall, the mice with pre-existing immune responses had higher IgG1 titers following vaccination than the naïve but vaccinated mice.

## 4. Discussion

Influenza A viruses are divided into two groups based on two proteins on the surface of the virus, with 18 different HA subtypes and 11 different NA subtypes [35,36]. Over 100 influenza A virus HA/NA combinations have been identified in wild bird populations, with additional reassorted viruses being possible [37]. However, while it is possible that the next pandemic could be initiated by a virus with any subtype, this report focused on three HA subtypes, H2, H5, and H7, since viruses of these subtypes continue to infect people, are highly lethal, and can have devasting effects on the poultry industry [38,39].

People infected with influenza viruses can elicit and imprint memory B cells that may be recalled upon vaccination with seasonal influenza vaccines [40]. However, the effect of pre-existing immunity to seasonal influenza viruses on vaccine-elicited immune responses by pre-pandemic HA vaccines has not been thoroughly explored [41]. The use of historical influenza viruses to infect mice or ferrets to establish pre-immunity prior to vaccination has successfully been used to test broadly reactive or universal influenza vaccines to mimic the pre-existing immunity in people [28,42,43]. Ferrets infected with an H1N1 A/Singapore/8/1986 virus and subsequently vaccinated with a single dose of a COBRA H1 HA elicited HAI activity against a broad panel of H1N1 influenza viruses [42,44]. Similar results were obtained using the same regimen with a historical H3N2 virus (A/Panama/2007/1999) to establish pre-immunity before vaccination [42,44]. The immune responses elicited by H1 or H3 COBRA HA vaccines were more robust following immunization in these pre-immune models [42,44]. Therefore, in this study, pre-immunity was established following the simultaneous infection of these historical H1N1 (group 1) and H3N2 (group 2) viruses prior to vaccination with pre-pandemic COBRA HA vaccines to determine its effect on the protective elicited immune responses.

The mice vaccinated with COBRA HA vaccines had higher HAI titers following vaccination in animals with pre-existing seasonal influenza A immunity than in immunologically naïve mice following the first vaccination. However, after the second vaccination, there was no significant difference between the two groups (Figure 4). Vaccine-elicited antibodies with HAI activity specifically bind to regions of HA that block HA receptor binding to host sialic acids to mediate entry [45]. Vaccinated mice that were pre-immune to seasonal influenza viruses also had higher anti-stem binding antibodies than naïve mice vaccinated with the same vaccines after the first vaccination (Figure 5). Therefore, COBRA HA vaccines elicit antibodies to both the head and stem regions of HA. Stem-directed antibodies can neutralize influenza virus infections by preventing the initial step of conformational change and inhibiting the membrane fusion activity of HA, thus inhibiting viral-endosomal membrane fusion during viral entry and causing the viral genome to be released into cytosol [46]. Following the second vaccination, HA head-based antibodies with HAI activity are immunodominant and are highly potent, leading to protection against viral infection. As observed previously, pre-pandemic COBRA HA vaccines elicit a wide range of HAI activity that does not always correlate with the traditional 1:40 protective HAI titer observed as a benchmark for human vaccinations [27,28,33,47,48]. Therefore, it is possible that antibodies directed against HA may have additional effector functions other than HAI activity. While stem-directed antibodies are broader, but less potent, they can work through multiple mechanisms, such as antibody-dependent cell-mediated cytotoxicity (ADCC), which may or may not correlate with protection [49]. Following natural infection, HA stem-directed antibodies are detected, but are not definitively associated with protection against seasonal influenza virus infection. However, it is yet to be determined if HA stem-directed antibodies that are elicited following seasonal influenza virus infection are sufficient to protect people against future pre-pandemic strains [50]. Higher baseline HA stem antibodies can reduce the duration of shedding, but not necessarily the severity of the disease, since some individuals still develop significant illness following an influenza virus infection [50].

COBRA rHA vaccines elicited the robust Th2-type, and significantly higher IgG1 responses following vaccination (Appendix A). In contrast, mice with pre-existing immunity induced by viral infection, and then vaccinated with COBRA HA vaccines, had more balanced IgG1/IgG2a responses (Figure 6) that are indicative of a more mixed T helper cell response in triggering effector mechanisms [51]. In this study, the COBRA HA antigens were formulated with AddaVax^TM^, a squalene-based MF-59-like adjuvant [52,53]. This adjuvant induces predominantly Th2-type humoral responses, preferentially stimulates CD4+ T cells to help B cell differentiation into IgG1-secreting cells in the presence of IL-4 and other anti-inflammatory cytokines by attracting more immune cells to site of injection, and upregulates antigen trafficking and presentation [54,55]. Intranasal infection with influenza viruses results in a more Th1-biased immune response, as observed in pre-immune, but mock-vaccinated mice (Figure 6). Mice vaccinated with COBRA rHA vaccines had higher IgG1 titers, and therefore overcame the virus-induced Th1 bias following infection. IgG1 antibodies are more effective in neutralizing activity, whereas IgG2a antibodies efficiently clear a viral infection from the host cells [56,57]. Following vaccination, the higher affinity to FcRs and the robust induction of IgG1 and IgG3 effector functions can lead to strong immune responses in human [51,58]. With or without pre-immunity, COBRA vaccinations induced high titers of both IgG1 and IgG2a antibodies, but these same vaccines elicited primarily IgG2a responses in pre-immunized mice, thereby reducing viral titers and disease symptoms. Following vaccination, IgG1 contributed to the vast majority of the protection by HAI activity, and it most likely overwhelmed the IgG2a antibodies in response to pre-immunization.

Regardless of the subtype, COBRA HA vaccines elicited antibodies with HAI activity against a broad number of divergent and future-drifted influenza virus strains within a subtype [22,26,27,28,29,30,31,32,33,34,42,43,44,59]. The H5 COBRA HA (IAN8) component of this trivalent vaccine elicited high HAI antibody titers against a future strain, A/Astrakhan/32112/2020 (H5N8), from the newly emerged highly pathogenic clade 2.3.4.4b [18]. A novel HPAI H5N8 virus in clade 2.3.4.4b was first identified in Eastern China in 2013 and quickly spread to Japan and South Korea through migrating birds [60,61]. Additional outbreaks were widely reported in some other countries in East Asia and Europe since the beginning of 2020 [62,63,64]. In December 2020, the first human H5N8 infection was reported when seven people were infected on a poultry farm in Russia, and the first U.S. case of human H5N1 infection was reported in Colorado in April, 2022 [18,65]. Currently, 46 states in the U.S. are experiencing a devasting H5Nx outbreak in the poultry industry, with 52.8 million chickens slaughtered in 2022 [66]. The H5 COBRA HA-elicited antibodies had HAI activity against H5Nx viruses from multiple clades isolated between 2011 and 2021, reinforcing the finding that IAN8 HA is an effective vaccine for both current and future avian and human infections [33].

Live poultry, especially in live bird markets, are a significant channel for zoonotic influenza viruses to jump and spread to humans [67]. Eighty-two percent of human infections during the first H7N9 outbreak in China in 2013 were the result of exposure to infected poultry at a live bird market [68]. Consistent surveillance and monitoring of these markets should be prioritized to reduce avian-to-human influenza virus infections. The H7 component in the COBRA trivalent vaccine induced antibodies with a wide breadth of HAI activity against both North American and Eurasian H7 HA lineages that are effective against multiple H7 strains.

H2Nx viruses currently circulate in birds and occasionally spread to swine farms, which can result in morbidity and mortality in pigs, as well as farm handlers of these infected animals. In 2006, the H2N3 isolate, A/Swine/Missouri/2124514/2006 (SW/MO/2006), was isolated from 6-week-old pigs on two farms in the U.S. This virus was able to efficiently transmit to ferrets and cause severe disease [25]. The H2 COBRA HA vaccine elicited HAI activity against viruses from both clades of H2Nx that are circulating today, including SW/MO/2006, as well as the historical H2N2 viruses that caused the 1957 pandemic and are now only located in research laboratories [22,27,28]. H2 viruses could transmit from a zoonotic species and reassort with a human virus, which most likely happened in 1957 and again in 1968. Today, few people under 50 years of age have no pre-existing immunity to H2N2 viruses. Laboratory personnel could benefit from the H2 COBRA HA vaccine since laboratory-acquired infections have occurred previously and it has been speculated that these types of lab infections may have led to influenza-related outbreaks [10,22]. Vaccinating laboratory workers, if not the general population, with these H2 COBRA HA vaccines could set up long-lasting memory B cells and antibodies that could be recalled against future H2Nx pandemics [27,28].

Taken together, precisely predicting the subtype, clade, or specific strain of the influenza virus that will cause the next pandemic, including when and where it will emerge, is currently not impossible. A multivalent vaccine that incorporates several influenza subtypes of pre-pandemic risk to elicit broadly reactive antibodies would be highly valuable [69]. The COBRA HA trivalent pre-pandemic vaccine used in this study incorporated H2, H5, and H7 soluble HA antigens and can be used as a vaccine candidate to reduce infection and spread in domestic poultry or as a pre-pandemic vaccine for any future human outbreak.

## Figures and Tables

**Figure 1 viruses-15-00203-f001:**
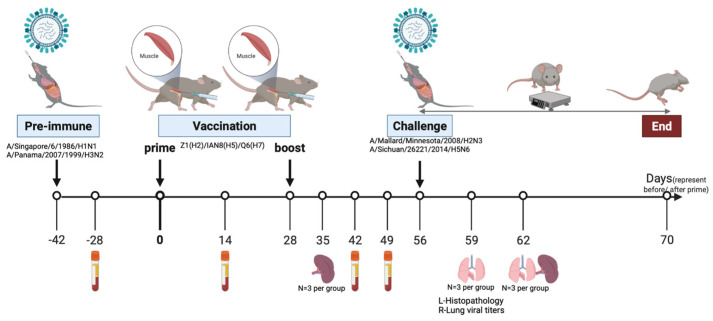
Study design. Schematic diagram of the pre-immunization, vaccination, viral challenge, and sampling timeline. Eight-week-old female DBA2/J mice were inoculated intranasally with equal concentrations of A/Singapore/6/1986/H1N1 and A/Panama/2007/1999/H3N2 viruses or mock pre-immunized with PBS. Mice were vaccinated with trivalent COBRA HA vaccine consisting of Z1 (H2), IAN8 (H5), Q6 (H7) twice or mock-vaccinated with PBS six week after pre-immunization. Four weeks following boost, all the mice were intranasally challenged with A/Mallard/Minnesota/2008/H2N3, or A/Sichuan/26221/2014/H5N6. All mice were weighed and monitored for clinical symptoms from day 56 to day 70. Spleen samples were collected one week following boost. Blood was collected two weeks following pre-immunization and boost. On day 59 and 63 post-challenge, lung samples were collected.

**Figure 2 viruses-15-00203-f002:**
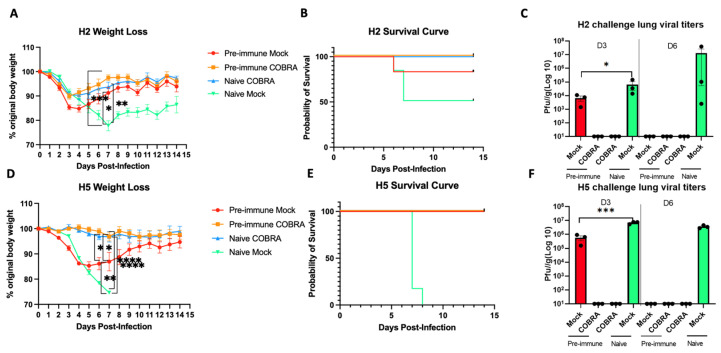
COBRA HA vaccinations confer effective protection against H2/H5 challenge. DBA2/J mice were intranasally challenged with A/Mallard/Minnesota/2008/H2N3 (**A**–**C**), or A/Sichuan/26221/2014/H5N6 (**D**–**F**). Mice were humanely euthanized 3 days and 6 days post-challenge for lung viral titers or monitored for 14 days post-challenge for survival record. Weight (**A**,**D**), survival (**B**,**E**), and lung viral titer (**C,F**) are measured for vaccine efficacy. Data shown are mean ± SEM representative of 2 independent experiments (A–C for H2, C–F for H5 challenge); n = 6 per group for weight change and survival, and n = 3 per group on both 3 days and 6 days post-challenge; * *p* < 0.05, ** *p* < 0.01, *** *p* < 0.001, **** *p* < 0.0001 (one-way ANOVA with Turkey’s post-test in panels **A**,**C**,**D**,**F**).

**Figure 3 viruses-15-00203-f003:**
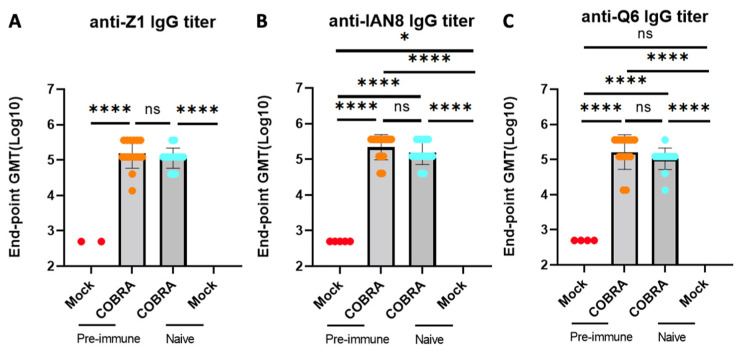
COBRA HA vaccinations induce high anti-H2/H5/H7 HA IgG antibodies. Serum from DBA2/J mice collected 2 and 3 weeks after boost vaccination was used to determine anti-Z1 (**A**), anti-IAN8 (**B**), and anti-Q6 (**C**) endpoint titer in each group. Data shown are mean ± SEM; n = 15 per group in each experiment; * *p* < 0.05, **** *p* < 0.0001. ns, not significant, *p* > 0.05 (one-way ANOVA with Turkey’s post-test).

**Figure 4 viruses-15-00203-f004:**
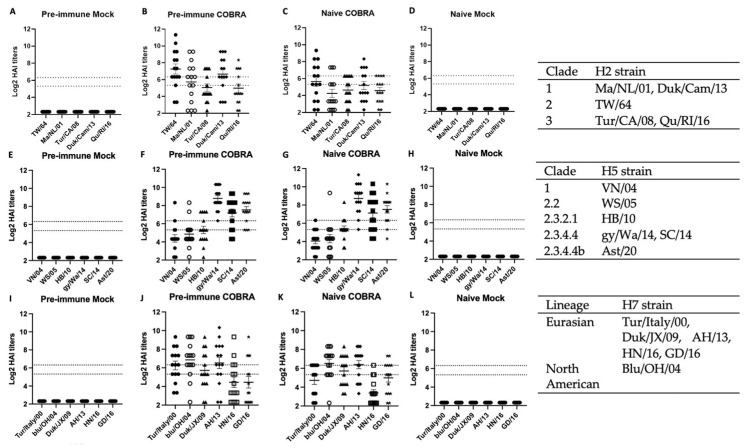
COBRA HA vaccinations can elicit broad hemagglutination-inhibition antibodies against panels of H2, H5, and H7 strains. Serum from mice of four different groups taken 2 and 3 weeks following final vaccination in a prime-boost model was tested for HAI activities. The two dashed lines represent HAI titers of 40 and 80. Row top (**A**–**D**; n = 15), middle (**E**–**H**; n = 15), bottom (**I**–**L**; n = 15) represent HAI panel of H2, H5, and H7, respectively. Each strain and the clade or lineage are located on the right side of each row. The H2 VLP panel is composed of Clade 1 HAs (A/Mallard/Netherlands/13/2001/H2N9 or Ma/NL/01 in hollow circle; A/Duck/Cambodia/419W12M3/2013/H2N2 or Dk/Cam/13 in diamond), Clade 2 HA (A/Taiwan/1/1964/H2N2 or T/64 in black circle), Clade 3 HA proteins (A/Turkey/California/1797/2008/H2N8 or Tur/CA/08 in triangle, A/Quail/Rhode Island/16-018622-1/2006/H2N2 or Qu/RI/16 in star). The H5 virus panel is composed of Clade 1 HA (A/Vietnam/1203/2004/H5N1 or VN/04 in black circle), Clade 2.2 HA (A/Whopper swan/Mongolia/244/2005 or WS/05 in hollow circle), Clade 2.3.2.1 HA (A/Hubei/1/2010/H5N1 or HB/10 in triangle), Clade 2.3.4.4 (A/Sichuan/26221/2014/H5N6 or SC/14 in square, A/gyrfalcon/Washington/41088-6/2014/H5N8 or gy/Wa/14 in diamond, A/Astrakhan/32112/2020/H5N8 or Ast/20 in star). The H7 VLP panel is composed of North American lineage HA (A/bluewing teal/Ohio/658/2004/H7N3 or blu/OH/04 in hollow circle), Eurasian lineage HAs (A/Turkey/Italy/589/2000/H7N1 or Tur/Italy/00 in black circle, A/Duck/Jiangxi/3230/2009 or Duk/JX/09 in triangle, A/Anhui/1/2013/H7N9 or AH/13 in diamond, A/Hunan/2650/2016/H7N9 or HN/16 in hollow square, A/Guangdong/17SF003/2016/H7N9 or GD/16 in star).

**Figure 5 viruses-15-00203-f005:**
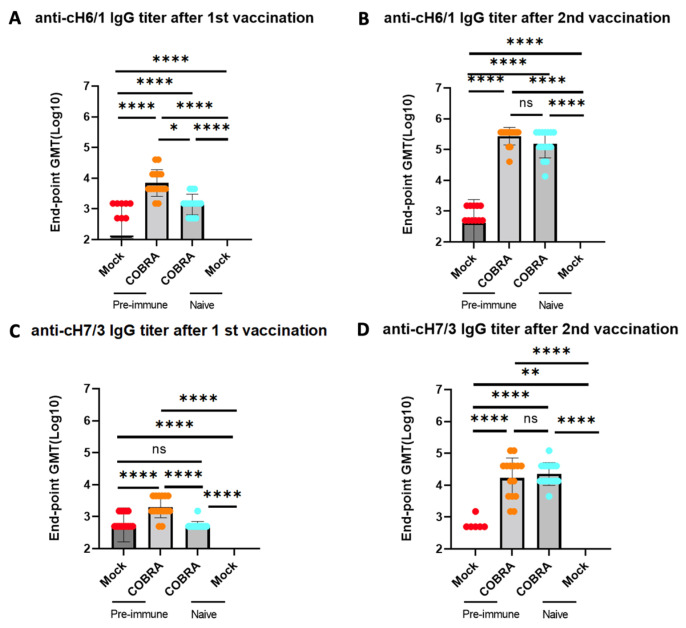
H1/H3 pre-immunity contributes to group 1 and group 2 HA stalk IgG antibody but shows little effect after two COBRA HA vaccinations. Stalk-reactive antibodies were detected by measuring binding of sera from DBA2/J mice collected 1 week after prime (**A**,**C**) and after boost (**B**,**D**) vaccination were used to determine anti-cH6/1 (**A**,**B**) and anti-cH7/3 (**C**,**D**) endpoint titer in each group. Chimeric HA cH6/1 bears the H6 head from A/Mallard/Sweden/81/2002 and H1 stalk from A/California/07/2009, and cH7/3 bears the H7 head from A/Anhui/1/2013 and H3 stalk from A/Texas/50/2012. Data shown are mean ± SEM; n = 15 per group in each experiment; * *p* < 0.05, ** *p* < 0.01, **** *p* < 0.0001. ns, not significant, *p* > 0.05 (one-way ANOVA with Turkey’s post-test).

**Figure 6 viruses-15-00203-f006:**
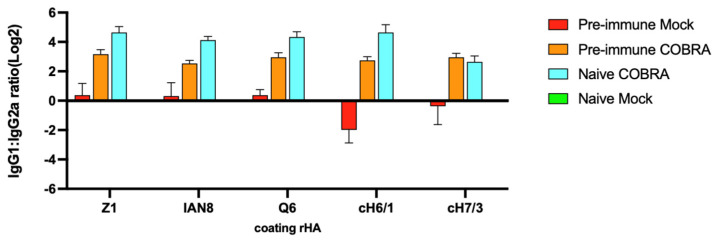
COBRA HA vaccinations elicit Th2-biased antibody responses, and pre-immunization elicits Th1-biased antibody response. Serum from DBA2/J mice collected 2 and 3 weeks after boost vaccination was used to determine anti-Z1, anti-IAN8, anti-Q6, anti-cH6/1, and anti-cH7/3 endpoint titer in each group. IgG1:IgG2a ratio is shown. Data shown are mean ± SEM; n = 15 per group in each experiment.

**Table 1 viruses-15-00203-t001:** Vaccination and infection schema of naïve and pre-immune mice.

Group	Pre-Immune	Dosage	Vaccination	Challenge	Dosage
1. Pre-imune Mock	Sing/86, Pan/99	5 × 10^6^ PFU/50 μL	Mock/PBS	Mal/MN/2008	2 × 10^7^ PFU/50 μL
2. Pre-immune COBRA	Sing/86, Pan/99	5 × 10^6^ PFU/50 μL	Z1/IAN8/Q6	Mal/MN/2008	2 × 10^7^ PFU/50 μL
3. Naive COBRA	Mock/PBS	50 μL	Z1/IAN8/Q6	Mal/MN/2008	2 × 10^7^ PFU/50 μL
4. Naive Mock	Mock/PBS	50 μL	Mock/PBS	Mal/MN/2008	2 × 10^7^ PFU/50 μL
5. Pre-imune Mock	Sing/86, an/99	5 × 10^6^ PFU/50 μL	Mock/PBS	Sichuan/2014	1 × 10^6^ PFU/50 μL
6. Pre-immune COBRA	Sing/86, Pan/99	5 × 10^6^ PFU/50 μL	Z1/IAN8/Q6	Sichuan/2014	1 × 10^6^ PFU/50 μL
7. Naive COBRA	Mock/PBS	50 μL	Z1/IAN8/Q6	Sichuan/2014	1 × 10^6^ PFU/50 μL
8. Naive Mock	Mock/PBS	50 μL	Mock/PBS	Sichuan/2014	1 × 10^6^ PFU/50 μL

## Data Availability

The data are contained within the article and Appendix A.

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
