# Peer review of "Evaluation of Pre-Pandemic Trivalent COBRA HA Vaccine in Mice Pre-Immune to Historical H1N1 and H3N2 Influenza Viruses"

_viruses, 2023, doi:10.3390/v15010203_

Round 1
Reviewer 1 Report
Comments for the authors of Viruses manuscript number viruses-2126212:
The authors of Viruses manuscript “Evaluation of pre-pandemic trivalent COBRA HA vaccine in mice pre-immune to historical H1N1 and H3N2 influenza viruses”, present their recent findings computationally optimized broadly reactive antigen (COBRA) vaccines against influenza viruses. In this study, they are testing to potential for pre-existing immunity against seasonal influenza A virus hemagglutinins (H1 and H3) to interfere with COBRA vaccines against H2, H5, and H7 hemagglutinins. Their results show that pre-existing immunity did not interfere with the ability for their pre-pandemic vaccines to induce antibodies. The authors demonstrate this using vaccine:challenge models, serological evaluation that includes ELISA (IgG), HAI assays, and ELISA against stalk domains. They also show that the COBRA vaccine-induced response is biased toward Th2. The study is well-designed and it helps address an important question related to real-world application of this vaccine. Importantly, showing that the pre-immunity does not inhibit the induction of stalk domain antibodies, coupled with the challenge data presented, strongly supports the conclusion that pre-existing immunity doesn’t interfere with COBRA vaccine-induced immunity. I only have one minor comment that I would like the authors to consider as they revise the manuscript.
General Comments:
- In the abstract, on line 21, the sentence that starts “Mice were fully protected mice from lethal challenge…” should be revised.
Author Response
Comment: In the abstract, on line 21, the sentence that starts “Mice were fully protected mice from lethal challenge…” should be revised.
Response: Thanks for pointing out the error of the sentence, it has been changed to “were fully protected from lethal challenge…”
Reviewer 2 Report
This study by Pan and Ted evaluated the effectiveness of trivalent COBRA HA vaccine in pre-immune (to human seasonal H1N1 and H3N2) mice. In most vaccine studies, the naïve animals are used to assess the effectiveness of the vaccine. But in this study, the authors used the pre-infected mice that partly mimics the human immune status. Naturally, almost all people were exposed to seasonal influenza viruses or vaccinated with different candidate vaccine viruses. The immune history (also known as ‘antigenic sin’) has an important role on the effect of the immune response to the infection or vaccination by current strains. About the H5Nx, H7N9 viruses, people are concerned if they can cause a pandemic because of sporadic transmission from avian to human and regard them as the pandemic potential viruses. So, the preparedness of the vaccine against these viruses is important and necessary for public health. The following are my questions about this study:
1. The authors designed the vaccine using the HA protein of H2, H5 and H7 subtypes, but only used the H2N3 and H5N6 as the challenge viruses. Could you explain why not use H7 virus
2. In result 3.4, the author tested the anti-HA stem antibody against the H1 and H3 HA stems. Is it better also to test the antibody against the H2, H5 and H7 HA stem?
Author Response
Comment #1: The authors designed the vaccine using the HA protein of H2, H5 and H7 subtypes, but only used the H2N3 and H5N6 as the challenge viruses. Could you explain why not use H7 virus
Response: Our group would like to use an H7 virus for challenge, however, we did not have an H7Nx virus to use at BSL2 level at the time of this study. We hope in the future to use a HPAI H7 virus to study protection against H7 infection. The current does not seem diminished without the H7 challenge of vaccinated mice. The immunological data supports the effectiveness of the vaccine in this study.
Comment #2: In result 3.4, the author tested the anti-HA stem antibody against the H1 and H3 HA stems. Is it better also to test the antibody against the H2, H5 and H7 HA stem?
Response: H1, H2 and H5 locate in phylogenetic group 1, they share similar stalk domain but distinct head domain, so chimeric proteins bearing mismatched head and stalk are widely used to test the anti-stalk antibody. Antibody against H1 HA stem induced can also be considered as anti-group 1 HA antibody. Same for H3 and H7, anti-H3 stalk antibody is also considered as anti-group 2 HA antibody. Also, , the chimeric proteins utilized in this study are among the most common and accessible ones. Overall, anti-HA stem antibody against H1 is considered the antibody against H2 and H5 stem, anti-HA stem antibody against H3 is considered the antibody against H7 HA stem.